# Effectiveness of Three Front-of-Pack Food Labels in Guiding Consumer Identification of Nutrients of Concern and Purchase Intentions in Kenya: A Randomized Controlled Trial

**DOI:** 10.3390/nu16223846

**Published:** 2024-11-10

**Authors:** Shukri F. Mohamed, Caroline H. Karugu, Samuel Iddi, Veronica Ojiambo, Caliph Kirui, Gershim Asiki

**Affiliations:** 1Chronic Disease Management Unit, African Population and Health Research Center (APHRC), Nairobi P.O. Box 10787-00100, Kenya; ckarugu@aphrc.org (C.H.K.); siddi@aphrc.org (S.I.); vojiambo@aphrc.org (V.O.); ckirui@aphrc.org (C.K.); gasiki@aphrc.org (G.A.); 2Department of Public and Occupational Health, Amsterdam Medical Centre, 1105 Amsterdam, The Netherlands; 3Department of Women’s and Children’s Health, Karolinska Institute, 17177 Stockholm, Sweden

**Keywords:** fat, front-of-pack-labels, Kenya, packaged foods, randomized controlled trial, salt, saturated fats, sugar, warning labels

## Abstract

**Background**: Front-of-pack-labels (FOPLs) on packaged foods provide essential information to help consumers make informed dietary choices. However, evidence on their effectiveness, particularly in low- and middle-income countries like Kenya, is limited. **Objective**: This study assessed the effectiveness of three FOPLs in helping consumers identify nutrients of concern in packaged food products and influencing their purchase intention in Kenya. **Methods**: A total of 2198 shoppers from supermarkets in Nairobi, Mombasa, Kisumu, and Garissa were randomized into three groups: Red and Green Octagon label (RG), Red and Green Octagon with icons (RGI), and Black Octagon Warning label (WL). In the control phase, participants were shown unlabeled images of packaged foods, followed by questions. In the experimental phase, the same images were presented with one assigned FOPL, and participants responded again to the same set of questions. Differences in correct identification of nutrients of concern and changes in purchase intention were analyzed using frequency tables and Chi-Square tests, while modified Poisson regression assessed FOPL effectiveness. **Results**: FOPLs significantly improved correct nutrient identification and reduced the intention to purchase unhealthy foods, with the WL proving most effective. **Conclusions**: These findings highlight the potential of FOPLs, particularly the WL, as an effective regulatory tool for promoting healthier food choices in Kenya.

## 1. Introduction

With rapid urbanization taking place in Kenya, obesogenic food environments are increasingly shifting consumers towards unhealthy food products, posing a critical public health challenge. The rising burden of diet-related non-communicable diseases (DR-NCDs) now accounts for 42.8% of all deaths in Kenya [1]. The prevalence of overweight and obesity has shown a concerning upward trend, rising from 25% in 2008/09 to 49% in 2022 [2,3]. This rapid increase is largely driven by changes in the food system, leading to poor dietary patterns dominated by energy-dense, nutrient-poor foods. These dietary shifts are major contributors to the rise in obesity and overweight [4], as well as to the growing incidence of non-communicable diseases (NCDs) such as cardiovascular diseases, type 2 diabetes, and certain types of cancer, which are now the leading causes of death in Kenya [5]. This escalation is largely attributed to the overconsumption of unhealthy diets, particularly pre-packaged food products high in fat, sugar, salt, and energy [6].

Front-of-pack-labels (FOPLs) are proposed as a strategy to improve dietary quality by providing simplified symbols on the front of packaged products, representing detailed nutrient declarations usually found on the back of food packaging [7]. This study uses theories from consumer behavior and health communication, which suggest that providing simplified, interpretive information on food packaging, such as FOPLs, can influence decision-making and encourage healthier choices [8,9]. By providing clear and accessible nutritional information, these labels aim to guide consumers toward healthier food choices. Additionally, FOPLs can serve as an incentive for manufacturers to produce healthier options and reformulate existing products to meet consumer demand [6]. Nutrient-specific FOPLs can be categorized into two types: interpretive labels, which provide nutritional information for guidance and an overall assessment of the product’s nutritional quality (e.g., traffic light system and warning labels (WLs), and non-interpretive labels, which present information without any specific judgment or recommendation (e.g., % GDA (Guideline Daily Amount) system) [10].

While systematic reviews suggest that FOPLs can enhance consumers’ product selection, improve knowledge, and aid in identifying healthier products, there is a notable gap in understanding their influence in contexts like Kenya [11,12,13,14]. Moreover, the implementation of FOPLs can vary between voluntary and mandatory schemes, with some countries opting for regulatory measures to ensure their adoption and standardization. Even though there is no research to show this, it is likely that mandatory schemes would likely lead to extensive manufacturer reformulation [15]. Research has shown an association between mandatory restaurant calorie labeling and reductions in body mass index (BMI), with areas implementing such regulations experiencing greater reductions in BMI compared to those without, suggesting that FOPLs might have similar effects [16].

The effectiveness of WLs and other FOPLs may vary depending on cultural contexts, literacy levels, and the design of the labels themselves. Among FOPLs, WLs have garnered attention as potentially impactful tools for highlighting products high in nutrients of concern (sugar, salt, or unhealthy fats) in low- and middle-income countries. For instance, countries like Chile [17], Peru [18], Mexico [19], and Uruguay [20] are currently using WLs, while South Africa has proposed their implementation to combat rising rates of obesity and related diseases [21,22].

The proposed research has significant practical implications for public health policy and the food industry in Kenya. The findings can inform current regulatory discussions surrounding the adoption of mandatory front-of-pack labeling systems as a tool to combat the growing burden of DR-NCDs [23]. Additionally, the results can guide food manufacturers in reformulating products to align with consumer demand for healthier food options, potentially leading to healthier dietary patterns at the population level [24]. However, there remains a significant knowledge gap in how these labeling strategies translate to the Kenyan context, where unique dietary habits, socioeconomic factors, and levels of health literacy may influence their effectiveness.

To address this gap, our study evaluates the effectiveness of various FOPLs and their potential influence on food choices in Kenya. Specifically, we investigate how well these labels guide consumer decisions regarding nutrients of concern and their impact on purchasing intentions. By comprehensively assessing the effectiveness of FOPLs, we aim to contribute valuable insights that can inform strategies and policies aimed at promoting healthier food choices in the context of rapid urbanization in Kenya. Through this research, we hope to lay the groundwork for the development of an FOPL standard tailored to the Kenyan context, ultimately fostering healthier dietary behaviors and mitigating the burden of DR-NCDs.

## 2. Materials and Methods

This study was a three-arm randomized controlled trial (RCT) conducted between November and December 2023. Participants were adults recruited from four counties in Kenya: Nairobi, Mombasa, Kisumu, and Garissa. We chose these counties because three of them are major cities in Kenya, and one is a township, allowing for a diverse geographic representation. The RCT assessed both within-subject and between-subject effects. The within-subject effect measured the difference between a product without FOPL and the same product with FOPL. The between-subject effect compared the differences among the three different FOPL conditions. This trial was pre-registered with the ISRCTN Registry (Registration ISRCTN82491256) [25].

### 2.1. Sampling Size and Sample Strategy

A minimum sample size of 2185 participants was calculated based on findings from a previous study [21], which reported the correct identification rate (relative risk 1.32) of unhealthy foods using WLs. Adjustments for various factors were made, including 80% power, a design effect size of 1.2, and a 10% non-response rate, to ensure statistical power and representation. A stratified sampling was used to allocate this sample size across the four selected counties (Nairobi, Mombasa, Kisumu, and Garissa), proportional to their respective population sizes. This approach resulted in a proportionate distribution of the sample size, with 1251 participants from Nairobi, 376 from Mombasa, 400 from Kisumu, and 172 from Garissa counties, thereby ensuring a representative sample across these diverse geographic areas.

### 2.2. Front-of-Pack-Labels (FOPLs) Tested

Three proposed FOPL symbols were tested in this study: Red and Green (RG), Red and Green with icons (RGI), and WL. These symbols were suggested by the Ministry of Health-led technical committee responsible for developing the Kenya Nutrient Profile Model (KNMP) and the FOPL standard. Figure 1 shows all three symbols, which are octagonal in shape.

In the RG label, nutrients of concern (salt, sugar, fat, and saturated fat) were written as text; red and green colors were assigned to denote if the nutrients were higher or lower than the unhealthy thresholds defined in the KNPM. Products with nutrients of concern exceeding the threshold were labeled with a red symbol, while those meeting or below the threshold were labeled with a green symbol. These symbols would appear on products if the nutrients of concern were present in the product.

The RGI label uses the same color code as RG labels. It additionally had abbreviated nutrient names (fat (F) and saturated fat (SF)) and pictorials (a spoon with a heap for sugar and a saltshaker for salt). Like the RG label, products exceeding the threshold for nutrients are designated with a red symbol, while those meeting or below the threshold receive a green symbol. Symbols would also appear if the nutrient of concern was in the product.

The WL. It is a black octagon that incorporates both text and images similar to RGI. Unlike the RG and RGI, these labels would only appear on food products that contain excessive or high levels of salt, sugar, total fats, and saturated fats, with the text “high-in” to denote thresholds higher than those set by the KNPM.

### 2.3. Recruitment and Eligibility of Study Participants

Study participants were recruited as they exited supermarkets, food shops, and kiosks in the selected counties. Participants eligible for inclusion in the study were individuals aged 18 years or older who frequently purchased packaged foods or drinks and were the main or shared the food purchasing decisions within their households. To ensure representation across diverse socio-demographic groups, participants were selected based on gender (male or female), age (18–29 or 30–50 years), education level (no education, primary, secondary, and post-secondary), income (low or middle-high), and residence (urban or rural). We excluded health professionals, tobacco industry employees, individuals working in the sugary drinks and food industry, professionals in the advertising sector, and employees of market research companies due to potential conflicts of interest or biases that these individuals might have.

Trained field interviewers, experienced in data collection, conducted participant recruitment and data collection. They received training on the study objectives, participant recruitment procedures, and questionnaire administration. After consent was obtained, data collection took place with eligible participants between November and December 2023.

### 2.4. Procedures

Participants were randomized to one of three FOPL symbols (RG, RGI, and WL; see Figure 1 for images of the labels) to examine whether the FOPL type influenced their ability to correctly identify the healthiness of food products (used as a marker of understanding) and whether the labels would influence their future intention to purchase unhealthy foods (used as a marker for potential effectiveness).

A manual process using an Excel sheet was used to randomly allocate study participants to one of three FOPLs. Randomization to the label type took place before participants were enrolled in the study, and both participants and field interviewers were unaware of the labels assigned to them. The initial randomization step involved using the sample allocation for counties and sub-counties to ensure an equal distribution of labels among participants in those specific counties. Participants were then assigned labels randomly based on their specific unique IDs. This process was then imported into the data collection tablets, and random symbols appeared for each participant ID during the interviews. As a result of this procedure, 33.6% (*n* = 738) of participants were exposed to the RG, 33.8% (*n* = 744) to the RGI, and 32.6% (*n* = 716) to the WL (Figure 2).

Each participant was exposed to both the control (images without labels) and experimental (images with labels) phases on the same day, with the aim of assessing the within- and between-subject effects. During the control phase, all participants viewed product images displayed on mock packages without any FOPL, and they responded to a set of questions. In the experimental phase, the same participants were randomly assigned to one of the three label conditions (intervention). They viewed the same product images seen in the control phase, but this time the product images were presented with a FOPL (the intervention), and they were asked to respond to an identical set of questions as in the control phase.

### 2.5. Stimuli

In this study, we used fictional images of both the single and paired products (See Appendix A). The single products were images of potato crisp, fruit juice, and soda, while the paired products were two packets of bread, yogurt, and breakfast cereals with distinct brand names. We created four sets of fictional products, encompassing all nine items: one set without FOPL as the control condition, and three sets with each having one of the following FOPLs: a red and green label, a red and green label with pictorials, or a WL. The labels were placed on the top right corner of each fictional food image.

Our choice of product categories was guided by common foods and beverages used in Kenya, with an aim to represent a mix of items often perceived as unhealthy (e.g., crisps and soda) and those with varying healthfulness (e.g., 100% fruit juice, bread, breakfast cereals, and yogurt). All participants were presented with the same product sets, with the only difference being the applied labels. Each product pair had one item with lower amounts of nutrients of concern (sugar, salt, fat, saturated fat).

### 2.6. Outcome Measures

For the single products (crisps, juice, and soda) assessment, the primary outcomes were whether the participant correctly identified the foods that were high in salt, sugar, and fat (yes, no, or do not know) and correctly identified the foods as unhealthy (healthy or unhealthy). All the single products that were used in this study were unhealthy. A product was considered high in nutrients of concern or unhealthy if it displayed one or more red-colored labels from either the RG or the RGI or one or more WLs.

In evaluating the paired foods (bread, yogurt, and breakfast cereal), the primary outcome was the participant’s ability to accurately identify the food product higher in salt, sugar, or fat and correctly identify the unhealthier food. For paired foods, a food product was considered higher in nutrients of concern or unhealthier if it featured one or more red labels (RG or RGI) or a WL.

We also assessed changes in intentions to purchase unhealthy food products using the question: “How likely are you to buy this product for yourself or your family?” Responses were recorded on a four-point Likert scale, with options including “I would definitely not buy it”, “I am unlikely to buy it”, “I will consider buying it”, and “I will definitely buy it”. For analysis, all responses were simplified into binary outcomes: 1 = yes, and 0 = no. The “yes” outcome combined responses “I will consider buying it” and “I will definitely buy it”, while the “no” outcome combined responses “I would definitely not buy it” and “I am unlikely to buy it”.

To determine the individual impact of each label, we compared the count of correct responses from study participants at baseline (when the products were displayed without a label) with the follow-up (i.e., when the product was displayed with one of the three labels).

### 2.7. Analysis

To analyze the within-subject effects, we conducted a comparison of the proportions of correct identification of high nutrients of concern and the changes in intention to purchase unhealthy foods before and after exposure to the FOPLs. Differences in these proportions were assessed using frequency tables and Chi-Square tests of association to determine significant variations in the correct identification of nutrients of concern across different FOPL symbols.

In the between-subjects analysis, we used a modified Poisson regression to assess the effectiveness of different FOPL symbols. The response variables focused on the correct identification of nutrients of concern and the overall perceptions of food healthiness. These binary response variables were analyzed using the Stata command for Generalized Linear Models (GLMs), specifying the Poisson family, the log-link function, and using the robust standard errors option. The exponentiated coefficients of the model provide the estimated relative risk ratio (RRR) rather than the odds ratio (OR). The main exposure variable was the three FOPL symbols, while covariates included the identification of nutrients of concern during the control phase (without symbols). The model was adjusted for sex and the role of being the decision-maker for food purchases in households since the other demographic factors were evenly distributed across the three arms. The results are presented as relative risk ratio (RRR) estimates comparing two distinct FOPLs. An RRR greater than 1 indicates that a higher percentage of participants exposed to one label correctly identified foods high in nutrients of concern or unhealthy products compared to those exposed to the other label. All analyses were performed using Stata version 15.

### 2.8. Ethics

Ethical guidelines were strictly adhered to throughout the implementation of this study. Participants gave informed consent to participate in the study before taking part. Ethical approval was obtained from the Ethics and Scientific Review Committee at AMREF Health Africa in Kenya (ERC/P1323/2022).

## 3. Results

A total of 2198 individuals participated in the randomized controlled trial and were included in the analyses. Table 1 displays demographic data categorized by the three FOPL conditions to which participants were randomized. Study participants were randomized into three arms as follows: A RG (33.6%), a RGI (33.8%), or a WL (32.6%). All demographic factors were evenly distributed across the three arms, except for sex and being the main decision-maker for food purchases in the home. There was a higher percentage of men in the WL arm (53.1%) and women in the RGI arm (55%). A significant difference was also observed in the category of the main decision-maker, with a higher percentage of participants identified as the main decision-makers in the RGI (79.4%) and WL (77.5%) arms compared to the RG arm (73.0%).

### 3.1. Identification of Nutrients of Concern of the Products and Unhealthiness Perception of Foods (Within-Subject Comparisons)

Figure 3 and Table A1 show the proportion of participants correctly identifying nutrients of concern in food products before and after FOPL exposure. The results demonstrate that exposure to FOPLs significantly improved participants’ ability to correctly identify nutrients of concern in most of the food products. Specifically, participants exposed to WL showed better identification of nutrients of concern and perceived the overall product unhealthiness more accurately across different food categories, including potato crisps, packaged juice, and Zanita soda. When comparing paired products with different nutritional content, FOPLs like RG and RGI were effective at identifying specific nutrients such as fats and sugar in breads, while the WL performed better at identifying salt and overall product unhealthiness.

### 3.2. Effectiveness of FOPLs in Identifying Nutrients of Concern (Between-Subject Comparison)

Modified Poisson regression analysis was used to compare the participants’ ability to correctly identify high levels of nutrients of concern in various food items using different FOPLs (Table 2). The statistical significance levels are denoted by asterisks (* *p* < 0.05). We compared WL against RG labels, WL against RGI, and RG against RGI for each food product.

When comparing exposure to WL versus the RG label, the WL was better at identifying breakfast cereals high in sugar compared to the RG label (RRR = 1.39, 95% CI: 1.20–1.61). The RG label was better at only identifying high sugar in yogurt compared to the WL (RRR = 0.80, 95% CI: 0.67–0.94). When comparing the WL versus the RGI label, the WL was better in identifying potato crisps high in salt (RRR = 1.10, 95% CI: 1.06–1.14), potato crisps high in fat (RRR = 1.10, 95% CI: 1.06–1.14), packed juices high in sugar (RRR = 1.10, 95% CI: 1.07–1.14), soda high in sugar (RRR = 1.08, 95% CI: 1.05–1.10), yoghurt high in sugar (RRR = 1.10, 95% CI: 1.09–1.14) and fat (RRR = 1.04, 95% CI: 1.01–1.06), breakfast cereal high in sugar (RRR = 1.50, 95% CI: 1.28–1.74) and fat (RRR = 1.03, 95% CI: 1.00–1.06), compared to the RGI label. The RGI was better at correctly identifying bread high in sugar (RRR = 0.70, 95% CI: 0.62–0.79) and fats (RRR = 0.71, 95% CI: 0.63–0.80) compared to the WL. When comparing the RGI versus the RG label, the RGI correctly identified bread high in sugar (RRR = 1.47, 95% CI: 1.29–1.67) and fats (RRR = 1.42, 95% CI: 1.25–1.62), while the RG correctly identified potato crisps high in salt (RRR = 0.89, 95% CI: 0.85–0.94) and fats (RRR = 0.90, 95% CI: 0.84–0.97), packaged juice high in sugar (RRR = 0.92, 95% CI: 0.88–0.96), soda high in sugar (RRR = 0.93, 95% CI: 0.88–0.97), bread high in salt (RRR = 0.95, 95% CI: 0.89–1.00), yoghurt high in sugar (RRR = 0.72, 95% CI: 0.60–0.86), and breakfast cereal high in sugar (RRR = 0.93, 95% CI: 0.92–0.95) and fats (RRR = 0.96, 95% CI: 0.89–1.02).

### 3.3. Effectiveness of FOPLs in Identifying the Overall Unhealthiness of Foods

The modified Poisson regression analysis was also used to compare the ability of the different FOPLs to correctly identify the overall unhealthiness of various food items. When comparing exposure to WL versus the RG label, the RG label was better at identifying the unhealthiness of packaged juice (RRR = 0.97, 95% CI: 0.95–0.99) and breakfast cereals (RRR = 0.26, 95% CI: 0.14–0.49) while the WL was better at identifying the overall unhealthiness of bread (RRR = 1.36, 95% CI: 1.05–1.76) and yoghurt (RRR = 2.26, 95% CI: 1.23–4.15). When comparing the WL versus the RGI label, the RGI label was better at identifying the overall unhealthiness in potato crisps (RRR = 0.90, 95% CI: 0.90–0.93), packaged juice (RRR = 0.87, 95% CI: 0.83–0.91), bread (RRR = 0.29, 95% CI: 0.15–0.55) and breakfast cereals (RRR = 0.25, 95% CI: 0.13–0.49). When comparing the RGI versus the RG label, the RGI label was better at correctly identifying the overall unhealthiness of potato crisps (RRR = 1.08, 95% CI: 1.00–1.17), packaged juice (RRR = 1.12, 95% CI: 1.05–1.18), soda (RRR = 1.03, 95% CI: 1.00–1.06), bread (RRR = 4.80, 95% CI: 2.13–10.80) and yoghurt (RRR = 2.40, 95% CI: 1.28–4.51).

### 3.4. Reduced Intention to Purchase Unhealthy Foods

Figure 4 provides insights into consumers’ intentions to purchase unhealthy food products based on different FOPL symbols. Overall, the findings suggest that the presence of labels generally reduced consumers’ intentions to purchase unhealthy foods compared to food products without labels. The RG and RGI labels had a similar effect in reducing consumers’ intentions to buy unhealthy foods, while the WL was the most effective in decreasing the intention to purchase all unhealthy food products compared to the other FOPLs.

## 4. Discussion

The results of this study provide valuable insights into the effectiveness of different FOPLs in improving consumers’ ability to identify nutrients of concern and their perception of the overall healthiness of food products in Kenya. Overall, exposure to FOPLs led to a significant improvement in participants’ ability to correctly identify nutrients of concern across various food categories, including potato crisps, packaged juice, soda, bread, yoghurt, and breakfast cereals compared to when the products had no FOPL on the food packaging. Findings from this study further showed that the presence of FOPLs enhanced consumers’ understanding of product healthiness and reduced consumers’ intentions to purchase unhealthy foods. Our results are consistent with existing evidence that shows that FOPLs are effective at helping consumers identify healthier choices [21,26]. Participants exposed to the WL demonstrated better identification of nutrients of concern and a reduced intention to purchase unhealthy foods compared to other FOPL symbols, such as RG and RGI. The RGI performed best in identifying unhealthy foods compared to the RG and WL.

Kenya developed two unique front-of-pack-labels (RG and RGI) that have not been used elsewhere, while the WL features for Kenya were adapted from the WL that is being proposed in South Africa [22] and similar WLs implemented across Latin American countries [27,28,29]. The labels used in the current study can be broadly categorized into two types: interpretive and non-interpretive. The RG and RGI labels are considered non-interpretive because they require more cognitive effort from the consumers to interpret the meaning of the red and green colors, where red indicates excess amounts of the nutrient of concern and green indicates that the nutrient is within or below threshold levels. In contrast, the WL is interpretive as it graphically communicates the product’s healthiness by explicitly stating “High in” for the nutrient of concern. This context is important as it provides a basis for interpreting our findings.

### 4.1. Identifying Unhealthy Foods

The results indicate that regardless of the label used, exposure to any of the FOPLs significantly enhanced participants’ ability to correctly identify nutrients of concern in most of the food products. This finding is consistent with a prior similar study conducted in South Africa [21], which also concluded that the presence of a front-of-pack-label on a product aided consumers in better identification of nutrients of concern in packaged foods compared to when the product lacked a FOPL. However, some participants who correctly identified nutrients of concern did not consistently interpret these as indicating the product was unhealthy, which explains the contradiction in the proportions of unhealthy foods identified compared to nutrients of concern. Similarly, another study conducted across 12 countries testing five FOPLs reported that the presence of FOPLs led to an improvement in the number of correct responses in the ranking task [30].

In the current study, the WL was the best at identifying nutrients of concern. The results of our study support findings from other contexts, indicating the widespread effectiveness of WLs as an effective regulatory measure. Several studies in different settings have found that WLs improved consumers’ ability to identify high levels of nutrients of concern in food products [21,26,31]. The RG and RGI labels use a color-coding system similar to the multiple traffic lights (MTLs) system, but with only two colors compared to the three colors used in the MTL system. Some participants may have struggled to connect the identification of nutrients of concern with the overall unhealthiness of the food, particularly when multiple labels and colors were used, leading to contradictory perceptions. The use of the green color is also associated with a health halo effect, and this could be misleading as consumers may perceive foods with green labels as healthy [15]. This perception could explain why consumers who saw a green label among the red labels may not have correctly identified the food product as unhealthy. Previous research has shown that consumers found the MTL challenging to interpret when multiple labels and colors required interpretation [32,33].

When analyzing within-subject effects, the WL performed the best in identifying the overall product unhealthiness in most products compared to the other two labels. However, in the regression analysis, the RGI label proved to be the most effective in helping participants correctly identify foods as unhealthy, outperforming both the WL and the RG label. This result highlights that even though the WL was more effective at pointing out nutrients of concern, many participants did not equate these nutrients with overall product unhealthiness. It is likely that the RGI label’s combination of color coding and icons seemed to confuse participants in judging the overall healthiness of the foods. Similar confusion in identifying unhealthy foods was noted in a study in Brazil using the TL system, showing that the presence of the different colors (green, amber) for nutrients of concern on the same product may have led participants to wrongly perceive the food product to be healthier than it was [26]. Therefore, using a color-coded system is likely to confuse consumers in identifying unhealthy products, thus reducing the intended effectiveness of the label to change consumers behavior.

### 4.2. Reducing Intention to Purchase Unhealthy Foods

Overall, the findings suggest that the presence of labels influenced purchasing intentions. The FOPLs generally reduced consumers’ intentions to purchase unhealthy foods compared to food products without labels. Although intention does not necessarily equate to actual purchasing, a shift in consumers’ intentions represents a crucial phase in the progression from exposure to front-of-pack-labels to real behavioral changes [34,35]. The current study investigated the impact of FOPLs on consumers’ intentions to purchase unhealthy food choices. The findings demonstrate that all three labels influenced participants’ reported intentions to buy unhealthy products, which is similar to what was reported in South Africa [21]. However, the WL was more effective in reducing the intention to purchase these unhealthy products than either the RG or the RGI labels. Our findings are consistent with several studies that have demonstrated the effectiveness of WLs in enhancing consumers’ understanding of product healthiness and influencing their purchasing decisions. A study by Taillile et al. [36] found that Chile’s implementation of WLs on unhealthy food products led to a significant decrease in purchases of these items. Similarly, a study by Roberto et al. showed that WLs were more effective than other FOPLs used in the study in reducing consumers’ intentions to purchase sugary beverages [37] in the US. Another study in Jamaica found that the WLs significantly outperformed the other FOPLs tested in helping consumers to choose to purchase the least harmful option [31]. South Africa also reported reduced intention to purchase unhealthy products when participants were exposed to a WL compared to the multiple traffic lights (MTLs) system or the GDA [21]. Similarly, a study by Khandapour et al. in Brazil demonstrated that WLs had a more significant substitution effect, leading consumers to shift their intentions away from purchasing unhealthy products towards opting for healthier alternatives [26]. In contrast to our findings, a study by Machín et al. found there was no difference between the effect of the WL and the traffic light label [33].

Our study contributes to the growing body of evidence supporting the role of WLs in addressing public health challenges related to diet-related diseases. By providing consumers with clear and easily understandable information about the nutritional content of food products, WLs empower individuals to make healthier choices and contribute to reducing the prevalence of obesity, diabetes, and other non-communicable diseases. These findings confirm consumer behavior theories that suggest the provision of simplified, interpretive information on food packaging can influence decision-making and promote healthier choices. Furthermore, the insights from this study can guide the development of evidence-based health policies aimed at promoting healthier food environments, enhance consumer education efforts to improve nutritional literacy, and inform the food industry’s strategies to reformulate products and adopt clearer labeling practices. These applications will be critical for mitigating the burden of diet-related non-communicable diseases and promoting population-level health and well-being.

### 4.3. Strengths and Limitations

This study has several strengths. With the large and diverse sample size of participants recruited from four counties in Kenya, including both rural and urban areas experiencing rapid urbanization, the study’s findings hold broader applicability to the Kenyan population. The randomized controlled trial (RCT) design minimized bias and facilitated comparisons across different FOPL conditions. By comprehensively evaluating three distinct FOPL symbols—RG, RGI, and WL—this research offers valuable insights into the effectiveness of various FOPLs in enhancing participants’ understanding of product healthiness and influencing their food choices. However, several limitations need to be considered. First, the study’s use of fictional images of food products may limit the generalizability of findings to real-world purchasing decisions, potentially affecting participants’ responses. Additionally, the cross-sectional design limits the evaluation of long-term effects. Lastly, the study prioritized FOPL formats recommended by the Kenyan Nutrient Profile Model (KNPM) technical committee to ensure relevance within the local regulatory framework, and while this limited the exploration of other well-known FOPL designs, it enabled a deeper assessment of labels likely to be implemented in Kenya, offering valuable insights for policymakers. Despite these limitations, the study’s rigorous methodology and comprehensive evaluation of FOPLs contribute valuable insights into the potential impact of FOPLs on food choices in Kenya.

### 4.4. Recommendation

We recommend that the Kenya Ministry of Health (MOH) implements WLs on a mandatory basis for all packaged foods and beverages to improve population health and reduce the diet-related NCD burden. Mandatory labeling can create stronger incentives for the industry to reformulate their products, as evidence indicates that voluntary schemes are less likely to achieve the intended outcomes of FOPLs, such as influencing consumer behavior and encouraging manufacturers to improve product formulations [38]. Mandatory WL would be particularly beneficial in settings with low nutritional literacy, such as Kenya, where they can help consumers make more informed choices despite limited knowledge about nutrition. Implementing WLs on food packaging can therefore empower consumers to make informed decisions and ultimately contribute to improving public health outcomes. Future research will be needed to investigate the effectiveness of the selected FOPL in Kenya.

## 5. Conclusions

In conclusion, this study provides evidence supporting the use of FOPLs in improving consumers’ ability to identify nutrients of concern and their perception of the healthiness of food products in Kenya. WLs significantly outperformed the other FOPLs in the study, such as RG and RGI, in enhancing consumers’ understanding of product health and influencing their intentions to purchase food products. These findings underscore the potential of FOPLs and specifically the WLs as a regulatory tool to promote healthier food choices and combat the growing burden of diet-related diseases in Kenya.

## Figures and Tables

**Figure 1 nutrients-16-03846-f001:**
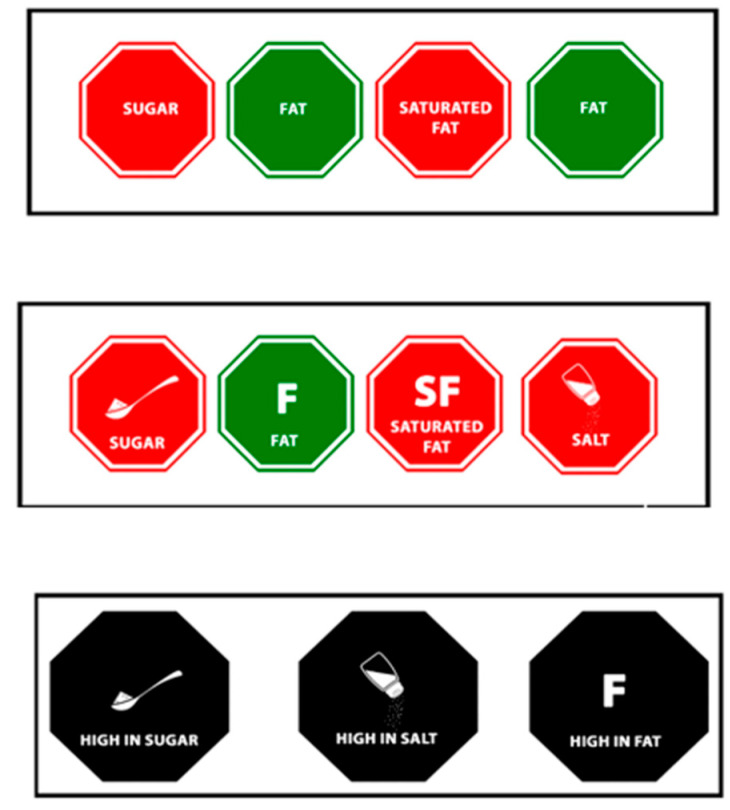
Three front-of-pack-labels tested in Kenya.

**Figure 2 nutrients-16-03846-f002:**
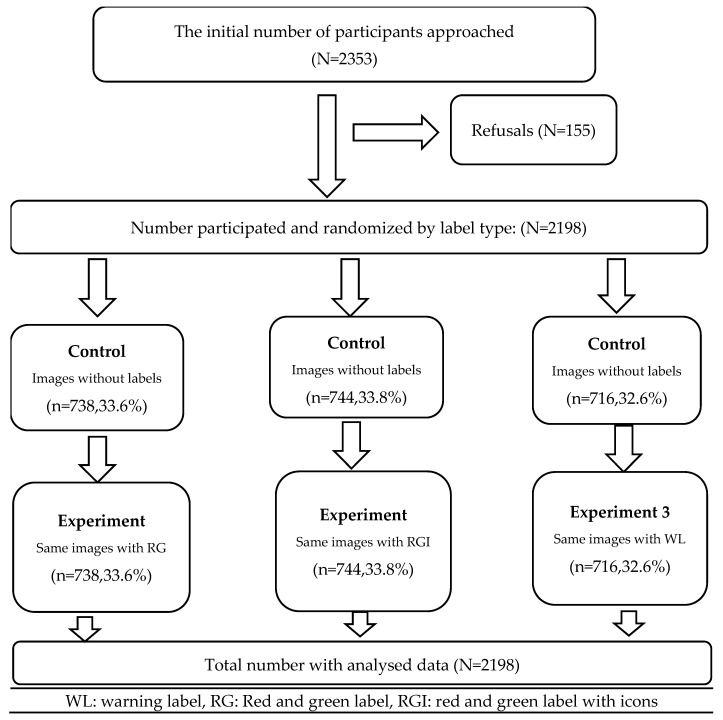
Study Flow Chart.

**Figure 3 nutrients-16-03846-f003:**
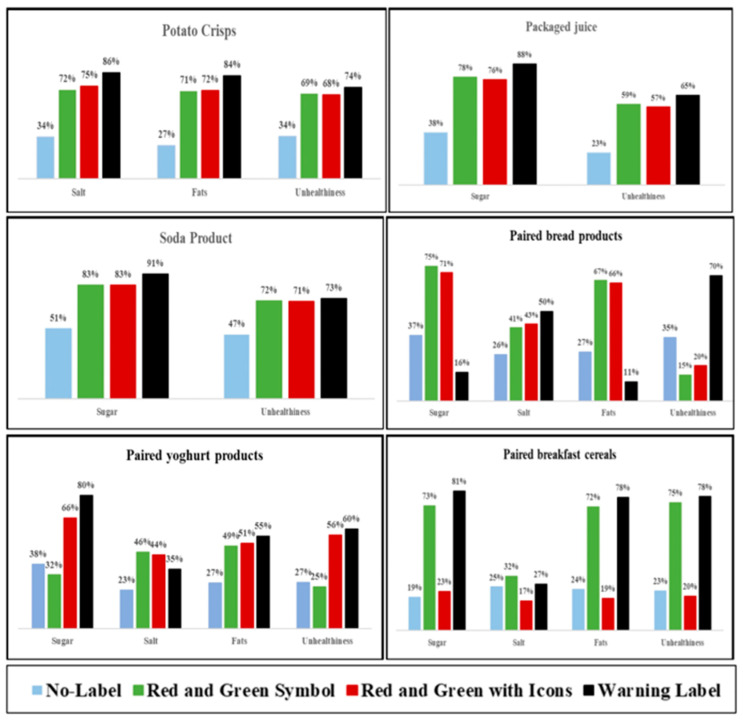
Proportions of correct identification of nutrients of concern and unhealthiness of products.

**Figure 4 nutrients-16-03846-f004:**
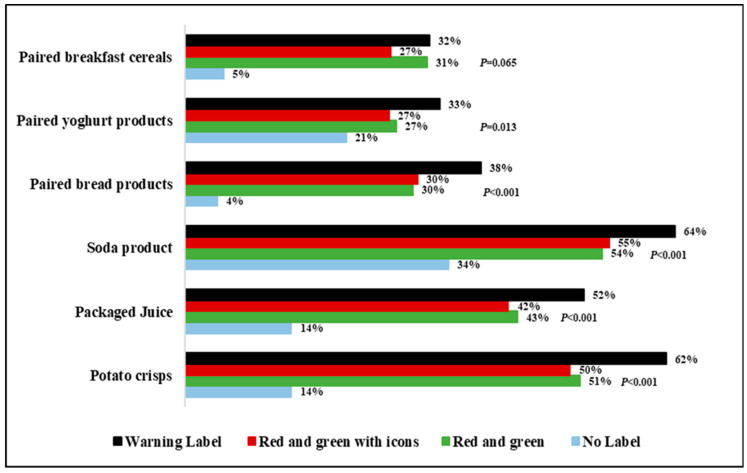
Reduced intention to purchase unhealthy foods by FOPL labels.

**Table 1 nutrients-16-03846-t001:** Participants’ socio-demographic information by FOPL symbols (N = 2198).

	RG (N = 738)	RGI (N = 744)	WL (N = 716)	Total (N = 2198)	*p*-Value
**County**					0.998
Nairobi	418 (56.6%)	430 (57.8%)	411 (57.4%)	1259 (57.3%)	
Mombasa	126 (17.1%)	126 (16.9%)	122 (17.0%)	374 (17.0%)	
Kisumu	132 (17.9%)	131 (17.6%)	128 (17.9%)	391 (17.8%)	
Garissa	62 (8.4%)	57 (7.7%)	55 (7.7%)	174 (7.9%)	
**Sex**					0.008
Male	363 (49.2%)	334 (44.9%)	380 (53.1%)	1077 (49.0%)	
Female	375 (50.8%)	409 (55.0%)	336 (46.9%)	1120 (51.0%)	
Intersex	0 (0.0%)	1 (0.1%)	0 (0.0%)	1(0.0%)	
**Age category**					0.751
18 to 29	268 (36.3%)	257 (34.5%)	247 (34.5%)	772 (35.1%)	
30 to 50	401 (54.3%)	427 (57.4%)	403 (56.3%)	1231 (56.0%)	
51 and above	69 (9.3%)	60 (8.1%)	66 (9.2%)	195 (8.9%)	
**Education level**					0.492
No education	0 (0.0%)	1 (0.1%)	2 (0.3%)	3 (0.1%)	
Primary school	151 (20.9%)	164 (22.6%)	136 (19.5%)	451 (21.0%)	
Secondary school	304 (42.2%)	298 (41.1%)	283 (40.5%)	885 (41.3%)	
Post-secondary	266 (36.9%)	262 (36.1%)	278 (39.8%)	806 (37.6%)	
**Marital Status**					0.131
Currently married	445 (60.3%)	431 (57.9%)	449 (62.7%)	1325 (60.3%)	
Previously married	72 (9.8%)	97 (13.0%)	81 (11.3%)	250 (11.4%)	
Never married	221 (29.9%)	216 (29.0%)	186 (26.0%)	623 (28.3%)	
**Employment status**					0.126
Formal employment	154 (20.9%)	183 (24.6%)	158 (22.1%)	495 (22.5%)	
Self-employed	304 (41.2%)	252 (33.9%)	261 (36.5%)	817 (37.2%)	
Casual workers	157 (21.3%)	167 (22.4%)	175 (24.4%)	499 (22.7%)	
Unemployed	105 (14.2%)	127 (17.1%)	105 (14.7%)	337 (15.3%)	
Farmers	13 (1.8%)	12 (1.6%)	16 (2.2%)	41 (1.9%)	
Others	5 (0.7%)	3 (0.4%)	1 (0.1%)	9 (0.4%)	
**Ethnic background**					0.976
Somali	62 (8.4%)	56 (7.5%)	56 (7.8%)	174 (7.9%)	
Luhya	79 (10.7%)	75 (10.1%)	83 (11.6%)	237 (10.8%)	
Luo	226 (30.6%)	220 (29.6%)	199 (27.8%)	645 (29.3%)	
Kikuyu	128 (17.3%)	127 (17.1%)	126 (17.6%)	381 (17.3%)	
Kamba	94 (12.7%)	101 (13.6%)	95 (13.3%)	290 (13.2%)	
Others	149 (20.2%)	165 (22.2%)	157 (21.9%)	471 (21.4%)	
**Parent with children < 18 years**					0.443
No	202 (27.4%)	194 (26.1%)	175 (24.4%)	571 (26.0%)	
Yes	536 (72.6%)	550 (73.9%)	541 (75.6%)	1627 (74.0%)	
**Main decision-maker**					0.012
No	199 (27.0%)	153 (20.6%)	161 (22.5%)	513 (23.3%)	
Yes	539 (73.0%)	591 (79.4%)	555 (77.5%)	1685 (76.7%)	

WL: warning label, RG: Red and green label, RGI: red and green label with icons, Chi-Square tests of association were used to determine significant variations.

**Table 2 nutrients-16-03846-t002:** Comparison of relative risk ratios for correct identification of nutrients of concern in various foods using different front-of-pack-labels (FOPLs).

Nutrients of Concern	WL vs. RG	WL vs. RGI	RGI vs. RG
	RRR(CI)	RRR(CI)	RRR(CI)
Potato crisp high in salt	0.98 (0.95–1.04)	1.10 (1.06–1.14) ***	0.89 (0.85–0.94) ***
Potato crisp high in fats	0.99 (0.95–1.03)	1.10 (1.06–1.14) ***	0.90 (0.84–0.97) **
Unhealthiness in potato crisps	0.99 (0.93–1.06)	0.90 (0.90–0.93) ***	1.08 (1.00–1.17) ***
Packaged juice is high in sugar	1.01 (0.99–1.04)	1.10 (1.07–1.14) ***	0.92 (0.88–0.96) ***
Unhealthiness of packaged juice	0.97 (0.95–0.99) **	0. 87 (0.83–0.91) ***	1.12 (1.05–1.18) ***
Soda is high in sugar	1.00 (0.93–1.06)	1.08 (1.05–1.10) ***	0.93 (0.88–0.97) ***
Unhealthiness of Soda	1.00 (0.95–1.04)	1.00 (0.95–1.03)	1.03 (1.00–1.06) *
Bread high in sugar	1.03 (0.98–1.07)	0.70 (0.62–0.79) ***	1.47 (1.29–1.67) ***
Bread high in salt	0.99 (0.95–1.02)	1.05 (0.97–1.14)	0.95 (0.89–1.00) *
Bread high in fat	1.01 (0.99–1.03)	0.71 (0.63–0.80) ***	1.42 (1.25–1.62) ***
Unhealthiness of Bread	1.36 (1.05–1.76) **	0.29 (0.15–0.55) ***	4.80 (2.13–10.80) ***
Yoghurt high in sugar	0.80 (0.67–0.94) **	1.11 (1.09–1.14) ***	0.72 (0.60–0.86) ***
Yoghurt high in salt	1.00 (0.97–1.04)	0.95 (0.85–1.05)	1.06 (0.93–1.21)
Yoghurt high in fats	0.99 (0.92–1.05)	1.04 (1.01–1.06) ***	0.95 (0.89–1.02)
Unhealthiness in yoghurts	2.26 (1.23–4.15) **	0.94 (0.89–0.99)	2.40 (1.28–4.51) **
Breakfast cereal high in sugar	1.39 (1.20–1.61) ***	1.50 (1.28–1.74) ***	0.93 (0.92–0.95) ***
Breakfast cereal high in salt	1.01 (0.97–1.04)	0.94 (0.85–1.05)	1.07 (0.94–1.21)
Breakfast cereal high in fats	0.99 (0.93–1.04)	1.03 (1.00–1.06) *	0.96 (0.89–1.02) *
Unhealthiness in breakfast cereals	0.26 (0.14–0.49) ***	0.25 (0.13–0.49) ***	1.04 (0.99–1.09)

WL: warning label, RG: Red and green label, RGI: red and green label with icons, RRR: Relative risk ratio, CI: Confidence Interval, * *p* < 0.05, ** *p* < 0.01, *** *p* < 0.001, Poisson regression was used to assess the effectiveness of different FOPL symbols, and the models are adjusted for sex and the role of being a decision-maker for food purchases in the household.

## Data Availability

Data analyzed for this paper form part of a primary project which is currently being written up in other publications. Data used for this paper will therefore be available upon request and granted for replication purposes. Anonymized data will be available from the African Population and Health Research Center (APHRC) Microdata portal (https://aphrc.org/microdata-portal/, accessed on 30 January 2026). For data inquiries, please contact Shukri Mohamed (smohamed@aphrc.org).

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
