# Peer review of "Effectiveness of Three Front-of-Pack Food Labels in Guiding Consumer Identification of Nutrients of Concern and Purchase Intentions in Kenya: A Randomized Controlled Trial"

_nutrients, 2024, doi:10.3390/nu16223846_

Round 1
Reviewer 1 Report
Comments and Suggestions for Authors
This paper is well written and organized. However, there are some corections before acceptance.
1. Since the research gap was mentioned in the intriduction, research questions should be presented accordingly. To answer these questions, the authors should address what they will study.
2. The theoretical and practical contributions in the intriduction are poor. Based on your key findings, these should also be discussed in depth in the discussion section.
3. In Table 1, only gender and decision maker groups were significant. Why was that?
Author Response
This paper is well written and organized. However, there are some corrections before acceptance.
Response: Thank you for taking time to review this manuscript and to provide valuable feedback that will improve it.
- Since the research gap was mentioned in the introduction, research questions should be presented accordingly. To answer these questions, the authors should address what they will study.
Response: Thank you for this comment. We have updated the text in the introduction section incorporating the research questions and how our current study will answer these questions. See lines 88-96.
- The theoretical and practical contributions in the introduction are poor. Based on your key findings, these should also be discussed in depth in the discussion section.
Response: Thank you for this comment. We have added practical and theoretical contributions of FOPLs in the introduction section. See lines 46-49 and lines 78-83. We have also linked our findings to these contributions in the discussion section. See 442-454.
- In Table 1, only gender and decision maker groups were significant. Why was that?
Response: Thank you for pointing out this observation. Although the randomization process was designed to balance the three FOPLs across all variables, the random nature of assignment led to significant differences in these two variables by chance. To address this, we adjusted for these variables in our analysis to minimize any potential confounding. This adjustment ensures that the observed effects are more likely to be attributed to the intervention itself, rather than the imbalances in these variables between the FOPL groups.
Reviewer 2 Report
Comments and Suggestions for Authors
Dear Authors,
This study was conducted to the effectiveness of three front-of-pack food labels in guiding consumer identification of nutrients of concern and purchase intentions in Kenya based on a randomized controlled trial. This manuscript has been well designed and written. I believe this was excellent issue in field of public health and nutrition section.
0. Abstract
Please reduce the number of characters in your abstract to 200 words or less based on MDPI guideline
Please delete abbreviation and sort alphabetically in Key-words.
1. Introduction
Line 46: Abbreviations are usually defined at the first use in the whole main text. ‘FOPLs’ à Front-of-pack-labels (FOPLs).
Please add sentences of introduce, background, or explain about obesity problems and disease from nutrition (food environments).
2. Method: well-describe
Line 92: front-of-pack label (FOPL) à FOPL
Line 181: Please add full name of abbreviations (FOPL, RG, RGI, and WL) in footnote.
Line 196, Line 198: Because abbreviations are already defined previously, please use abbreviations in whole manuscript. Moreover, I recommend that this manuscript should be edited by an English professional editor for more readable.
Front-of-Pack Labelling (FOPL) à FOPL; warning label (WL) à WL
3. Results
Line 265: In Table 1, please add full name of abbreviations (RG, RGI, and WL) and statistical methods in footnote.
Line 288: In Table 2, please add statistical methods in footnote.
5. Discussion
You should add application in field of this study.
6. Appendix B
Line 498: In Table A1, please add statistical methods in footnote.
Comments on the Quality of English LanguageModerate editing of English language required.
Author Response
Comments and Suggestions for Authors
Dear Authors,
This study was conducted to determine the effectiveness of three front-of-pack food labels in guiding consumer identification of nutrients of concern and purchase intentions in Kenya based on a randomized controlled trial. This manuscript has been well designed and written. I believe this was excellent issue in field of public health and nutrition.
Response: We thank the reviewer for taking the time to review this manuscript and to highlight the importance of this topic in public health and nutrition. Your comments are valuable and encourage us to continue contributing to this critical area of research.
We also appreciate the acknowledgment of the study's design and manuscript quality, as well as the importance of addressing front-of-pack labelling in guiding consumer behaviour.
- Abstract
Please reduce the number of characters in your abstract to 200 words or less based on MDPI guideline
Response: I have reduced the number of characters in the abstract to about 207 words.
Please delete abbreviations and sort alphabetically in Keywords.
Response: I have deleted the abbreviations and sorted the keywords alphabetically in the keywords. See lines 27-28.
- Introduction
Line 46: Abbreviations are usually defined at the first use in the whole main text. ‘FOPLs’ à front-of-pack-labels (FOPLs).
Response: Thank you for pointing this out. I have now defined "Front-of-Pack Label" (FOPL) in full on line 44, as this is the first mention in the manuscript, and I have used the abbreviation thereafter in accordance with standard practice.
Please add sentences of introduce, background, or explain about obesity problems and disease from nutrition (food environments).
Response: Thank you for this comment. We have added text in the background to discuss how obesity is linked to the food environment. See lines 36-41.
- Method: well-describe
Response: Thank you for your positive feedback on the description of the methodology. We are glad that the method section was clear and well-described.
Line 92: front-of-pack label (FOPL) à FOPL
Response: Thank you for this comment. I have deleted the full description of FOPL and left the abbreviation (FOPL).
Line 181: Please add full name of abbreviations (FOPL, RG, RGI, and WL) in footnote.
Response: This has been addressed. See line 192-193.
Line 196, Line 198: Because abbreviations are already defined previously, please use abbreviations in whole manuscript. Moreover, I recommend that this manuscript should be edited by an English professional editor for more readable.
Response: Thank you for your observation. We have carefully reviewed the manuscript and ensured that all abbreviations are used consistently after their initial full description. We have also had English language editing to enhance readability.
Front-of-Pack Labelling (FOPL) à FOPL; warning label (WL) à WL
Response: We have ensured that all abbreviations are used consistently after their initial full description.
- Results
Line 265: In Table 1, please add full name of abbreviations (RG, RGI, and WL) and statistical methods in footnote.
Response: Thank you for your suggestion. I have added the full names of the abbreviations (RG, RGI, and WL) and the statistical methods used as a footnote in Table 1. Please see lines [280-281] for the updated footnote text.
Line 288: In Table 2, please add statistical methods in footnote.
Response: Thank you for your comment. I have included the statistical methods as a footnote of Table 2. See lines 303-304.
- Discussion
You should add application in field of this study.
Response: I have revised a paragraph in the discussion section to bring out the applicability of the study. See lines 442-454.
- Appendix B
Line 498: In Table A1, please add statistical methods in footnote.
Response: Thank you for your suggestion. I have added the statistical method used for this table as a footnote. See line 526-527
Comments on the Quality of English Language
Moderate editing of English language required.
Response: Thank you for your comment. The manuscript has been thoroughly reviewed and edited to improve the quality of the English language.
Reviewer 3 Report
Comments and Suggestions for Authors
1. Figure 2 was too concise, It should include more content from the 2.4 section, such as line 182 to line 190 or more.
2. The title of Figure 4 was “Intention to purchase unhealthy foods by FOPL Labels”. However, it seems to be adequate to use the opposite title “Intention not to purchase unhealthy foods by FOPL Labels” or “reduced Intention to purchase unhealthy foods by FOPL Labels.”
3. Please report which statistical software was used in the manuscript. It is rare to find Poisson regression analysis in statistical software. Did the authors use log-linear models for Poisson distribution data? Or did the authors only directly use the relative risk ratio (RRR) to analyze the data?
4. Lines 290-300, please rewrite them, because most of them were not correct. For WL v.s. RG, the authors said that “the WL was better at identifying breakfast cereals high in sugar compared to the RG label. The RG label was better in only identifying high sugar in yoghurt compared to the WL.” However, there were six significant RRR, not two significant ones. For RGI v.s. RG, the authors said that “the RGI correctly identified bread high in sugar and fats.” However, there were seven significant RRR (> 1.0), not only one significant RRR (> 1.0). Please re-check them and fill the RRR in the corresponding contents, such as “the WL was better at identifying breakfast cereals high in sugar compared to the RG label (RRR = 1.39, p = ?).” For the p-value, please offer real p-values, not use “p < .05” expression (except for p < .001) for future meta-analysis reference. If necessary, section 3.3 may also need correction.
Author Response
Comments and Suggestions for Authors
- Figure 2 was too concise, It should include more content from the 2.4 section, such as line 182 to line 190 or more.
Response: Thank you for your comment. We have revised Figure 2 to include more content from section 2.4 as requested. The figure now provides more detail on the study design, illustrating that the same images were used for both the control and experimental phases, with the only difference being the addition of one of the three labels in the experimental phase, while the control images had no labels. See the revised figure 2 on lines 176-193.
- The title of Figure 4 was “Intention to purchase unhealthy foods by FOPL Labels”. However, it seems to be adequate to use the opposite title “Intention not to purchase unhealthy foods by FOPL Labels” or “reduced Intention to purchase unhealthy foods by FOPL Labels.”
Response: Thank you for suggesting a more clear title for figure 4. We have settled to use the following title: “Reduced intention to purchase unhealthy foods by FOPL labels.” We have therefore revised title 3.4 and the title for figure. See lines 339 and 348, respectively.
- Please report which statistical software was used in the manuscript. It is rare to find Poisson regression analysis in statistical software. Did the authors use log-linear models for Poisson distribution data? Or did the authors only directly use the relative risk ratio (RRR) to analyze the data?
Response: We have added a statement in the manuscript specifying the statistical software used for data analysis. See line 263. The following text has also been added to describe how Poisson regression was fitted using STATA. See lines 252-255: “These binary response variables were analysed using the Stata command for Generalized Linear Models (GLM), specifying the Poisson family, the log-link function, and using robust standard errors option. The exponentiated coefficients of the model provide the estimated relative risk ratio (RRR) rather than the odds ratio (OR)."
- Lines 290-300, please rewrite them, because most of them were not correct. For WL v.s. RG, the authors said that “the WL was better at identifying breakfast cereals high in sugar compared to the RG label. The RG label was better in only identifying high sugar in yoghurt compared to the WL.” However, there were six significant RRR, not two significant ones. For RGI v.s. RG, the authors said that “the RGI correctly identified bread high in sugar and fats.” However, there were seven significant RRR (> 1.0), not only one significant RRR (> 1.0). Please re-check them and fill the RRR in the corresponding contents, such as “the WL was better at identifying breakfast cereals high in sugar compared to the RG label (RRR = 1.39, p = ?).” For the p-value, please offer real p-values, not use “p < .05” expression (except for p < .001) for future meta-analysis reference. If necessary, section 3.3 may also need correction.
Response: Thank you for your detailed comment. Upon review, we acknowledge that there are indeed six significant associations when comparing the WL vs. RG, as you pointed out. We have clarified this in the manuscript by ensuring that Section 3.2 ("Effectiveness of FOPLs in identifying nutrients of concern") focuses on two significant associations related to nutrients of concern (yoghurt and breakfast cereal high in sugar), while Section 3.3 ("Effectiveness of FOPLs in identifying the overall unhealthiness of foods") highlights the remaining significant associations related to the overall unhealthiness of various foods (juice, bread, yoghurt, and breakfast cereal).
We have also added the respective RRR values and their 95% confidence intervals for each significant association in both sections (lines 305-323 and lines 325-338). We have chosen not to report exact p-values, as the 95% confidence intervals provide sufficient statistical information. Additionally, the footnote in Table 2 explains the asterisks denoting significance levels, making the interpretation clearer. We hope these adjustments address your concerns and improve the clarity of the results.
Round 2
Reviewer 3 Report
Comments and Suggestions for Authors
OK